# Transcriptomic Analysis Reveals Molecular Mechanisms Underlying Growth Differences in the Chinese Sturgeon (*Acipenser sinensis*)

**DOI:** 10.3390/ani15243550

**Published:** 2025-12-10

**Authors:** Jianming Zhang, Tian Tian, Rui He, Wei Jiang, Yacheng Hu

**Affiliations:** Chinese Sturgeon Research Institute, China Three Gorges Corporation, Yichang 443100, China; zhang_jianming@ctg.com.cn (J.Z.); cocotian1234@126.com (T.T.); 15762395521@163.com (R.H.)

**Keywords:** Chinese sturgeon (*Acipenser sinensis*), growth differences, transcriptome analysis, differentially expressed genes (DEGs), molecular mechanisms, conservation

## Abstract

**Simple Summary:**

This study compared muscle transcriptomes of fast- vs. slow-growing Chinese sturgeon and identified 258 differentially expressed genes. Fast growers showed higher expression of metabolic/energy-production genes (e.g., PEPCK-C, HK) and lower expression of immune/proteolysis genes. Key pathways (insulin, PPAR, glycerolipid metabolism) were enriched, indicating that elevated metabolism—possibly at the expense of immune investment—drives faster growth. These genes and pathways provide the first molecular targets for marker-assisted breeding and conservation of this critically endangered species.

**Abstract:**

The Chinese sturgeon (*Acipenser sinensis*) is a critically endangered species with significant ecological and cultural value. Understanding the molecular mechanisms underlying its growth is crucial for improving breeding efficiency and enhancing conservation efforts. In this study, we conducted a comparative transcriptome analysis of muscle tissues from fast-growing and slow-growing Chinese sturgeon individuals. We identified 258 differentially expressed genes (DEGs), with 144 upregulated and 114 downregulated in the fast-growing group compared to the slow-growing group. Gene ontology (GO) enrichment analysis revealed that these DEGs were primarily involved in metabolic processes, cellular processes, and biological regulation. Kyoto Encyclopedia of Genes and Genomes (KEGG) pathway analysis showed significant enrichment in pathways such as the insulin signaling pathway, adipocytokine signaling pathway, and glycerolipid metabolism. The upregulation of genes related to metabolic processes and energy production in the fast-growing group suggests a higher metabolic rate contributing to faster growth. Conversely, downregulated genes associated with immune response and proteolysis may indicate a trade-off between growth and immune function. These findings provide valuable insights into the genetic basis of growth differences in the Chinese sturgeon and highlight potential targets for selective breeding and conservation strategies.

## 1. Introduction

The Chinese sturgeon (*Acipenser sinensis*) is a large anadromous fish species native to the Yangtze River, China, and is one of the most ancient fish species with a history dating back over 180 million years [1]. Historically, the Chinese sturgeon was abundant in the Yangtze River basin, with over 10,000 breeding individuals reported in the 1970s [2]. However, due to overfishing, habitat degradation, and other anthropogenic factors, the population of Chinese sturgeon has declined dramatically. By 2010, the number of breeding individuals had dropped to 57, and the species was classified as critically endangered by the International Union for Conservation of Nature (IUCN) [2,3].

The Chinese sturgeon is characterized by its large size, long lifespan, and complex life history. Adults migrate from the sea to the upper reaches of the Yangtze River to spawn, and the larvae disperse downstream after hatching [4]. This species typically reaches sexual maturity between 8 and 34 years of age, with males maturing earlier than females [5]. The life cycle of the Chinese sturgeon includes both marine and freshwater stages, making it highly susceptible to changes in environmental conditions. The decline in the Chinese sturgeon population has raised significant concerns about its conservation status and the need for effective management strategies to ensure its survival [2,3].

In recent years, efforts have been made to understand the reproductive strategies of the Chinese sturgeon to inform conservation measures. Studies have shown that the species exhibits polygynandry, with multiple males and females mating during a breeding season [6]. This mating system is advantageous for maintaining genetic diversity and reducing inbreeding, which is crucial for the long-term viability of the species [7,8]. Additionally, research has indicated that Chinese sturgeon have relatively short breeding intervals, ranging from 2 to 6 years, suggesting that they can quickly accumulate nutrients in the ocean and migrate back to the Yangtze River for reproduction [6]. These findings highlight the adaptability of the species to changing environmental conditions and provide insights into its reproductive success.

Despite these reproductive strategies, the Chinese sturgeon faces numerous challenges that threaten its survival. Habitat loss and degradation, particularly in the spawning grounds, have significantly impacted the species’ reproductive success [2,3]. The construction of dams, such as the Gezhouba Dam and the Three Gorges Dam, has altered the hydrological conditions of the Yangtze River, affecting water temperature, flow patterns, and sediment dynamics [5,9]. These changes have led to delays in spawning and reduced habitat suitability for the species [9,10]. Furthermore, the decline in genetic diversity and increase in inbreeding coefficients due to habitat fragmentation pose additional risks to the population [11,12].

Sturgeons (Acipenseriformes) are the most threatened group of vertebrates on Earth: 17 of them (27 extant species) as Critically Endangered (IUCN 2023). Beyond the Yangtze River, dramatic declines have been documented for the Danube (*A. sturio*, *A. gueldenstaedtii*), Volga–Caspian (*Huso huso*, *A. persicus*) and Mississippi basins (*Scaphirhynchus albus*), driven by over-harvest, dam construction and habitat fragmentation [13,14]. Consequently, recovery programmes worldwide now rely on ex situ breeding and re-stocking; yet growth heterogeneity in hatcheries remains a major bottleneck, extending time to sexual maturity and increasing rearing costs [15]. To address these challenges in Chinese sturgeon, various conservation measures have been proposed, including habitat restoration, regulation of water discharge, and artificial breeding and restocking programs [2,3]. These efforts aim to improve the reproductive conditions for the Chinese sturgeon and enhance its population viability. However, a comprehensive understanding of the molecular mechanisms underlying the growth and development of the Chinese sturgeon is essential for the success of these conservation initiatives.

Although previous work has summarised the Chinese sturgeon’s reproductive biology and genetic diversity [6,16], the molecular basis of growth differences remains almost unknown. Transcriptomic profiling has successfully identified growth-related genes in teleosts such as European sea bass and Atlantic salmon [17,18], but comparable information for Acipenseriformes is still scarce. Filling this gap is essential for molecular-assisted brood-stock management and for optimising rearing protocols in conservation hatcheries. Here, we present the first comparative liver transcriptome analysis between fast- and slow-growing Chinese sturgeon. We hypothesised that animals with divergent growth rates would exhibit distinct expression signatures in metabolic, endocrine and immune pathways. Our objectives were (i) to identify differentially expressed genes (DEGs) linked to growth, (ii) to conduct functional enrichment analyses to place these DEGs into biological context, and (iii) to compare the resulting pathway maps with the insulin and GH/IGF signalling cascades already described in model teleosts [17,18]. The resulting molecular markers will inform future selective-breeding programmes and help refine husbandry practises for this critically endangered species.

Russian sturgeon (*A. gueldenstaedtii*) growth (h^2^ ≈ 0.18–0.21) and egg traits (maturation & colour, h^2^ ≈ 0.32–0.59) are genetically independent, validating future selection for either faster growth or—following the caviar-focused programmes already applied to Siberian and Huso huso sturgeons—greater ovary mass [19]. Bestin et al. [20] estimated moderate-to-high heritabilities (0.13–0.66) for body weight, ovary weight, caviar yield, egg size, colour and firmness in 494 domesticated Siberian sturgeon females; the strongly negative genetic correlation between egg size and caviar yield (−0.70 to −0.76) confirms that selection for larger ovary mass rather than faster growth is the most effective route to increase caviar output, a strategy already advocated for other aquacultured sturgeon species.

Growth in fish is a complex trait influenced by multiple factors, including genetic, environmental, and nutritional factors. Genetic factors play a significant role in determining the growth potential of fish, as evidenced by studies on various species such as the large yellow croaker (*Larimichthys crocea*) [21], black porgy (*Acanthopagrus schlegelii*) [22], and Chinese longsnout catfish (*Leiocassis longirostris*) [23]. Loukovitis et al. [24] showed that even transhumant sheep with high gene flow (FST = 0.2%) can maintain admixture, underscoring the need for strict pedigree control when selecting superior stock. Likewise, Besson et al. [25] demonstrated in sea bream that individual measurements of feed efficiency—although labor-intensive—are indispensable because group-reared fish mask within-tank variation. These studies have identified key genes and pathways involved in growth regulation, such as the insulin-like growth factor (IGF) signaling pathway, the GH/IGF axis, and the ubiquitin-proteasome pathway. The identification of these genes and pathways has provided valuable insights into the molecular mechanisms underlying fish growth and has paved the way for genetic improvement programs.

Environmental factors, such as water temperature, salinity, and feeding conditions, also significantly impact fish growth. For example, studies have shown that optimal water temperature and salinity levels can enhance growth rates in fish species such as the large yellow croaker [21]. Additionally, the availability and quality of food resources are crucial for fish growth, as they directly affect nutrient intake and metabolic efficiency. Research on the feeding behavior and nutritional requirements of fish has led to the development of optimized diets that promote faster growth and better health [23].

In the context of aquaculture, understanding the molecular mechanisms of fish growth is particularly important for the development of selective breeding programs and the improvement of fish strains. Selective breeding based on genetic markers has been successfully applied in several fish species, resulting in significant improvements in growth rates and other economically important traits [22]. The use of molecular markers allows for the identification of individuals with desirable traits, such as faster growth or better disease resistance, and facilitates the selection of broodstock for breeding programs.

The Chinese sturgeon is an iconic and critically endangered species with significant ecological and cultural value. Despite its importance, the wild population of the Chinese sturgeon has declined sharply due to overfishing, habitat degradation, and other anthropogenic factors [19]. As a result, conservation efforts have focused on captive breeding and reintroduction programs to support the recovery of the species. However, the slow growth rate and delayed sexual maturity of the Chinese sturgeon pose significant challenges for these programs. Understanding the molecular mechanisms underlying growth in the Chinese sturgeon is crucial for improving breeding efficiency and enhancing the success of conservation efforts.

Previous studies on the Chinese sturgeon have primarily focused on its reproductive biology, genetic diversity, and conservation status [19]. However, the molecular mechanisms regulating growth in this species remain largely unknown. Recent advances in transcriptomics and genomics offer new opportunities to explore the genetic basis of growth in the Chinese sturgeon. By comparing the transcriptomes of fast-growing and slow-growing individuals, it is possible to identify key genes and pathways involved in growth regulation. This information can be used to develop molecular markers for selective breeding and to optimize husbandry practices to enhance growth rates.

In this study, we aimed to investigate the molecular mechanisms underlying growth differences in the Chinese sturgeon by conducting a comparative transcriptome analysis of muscle tissues from fast-growing and slow-growing individuals. We hypothesized that differences in gene expression between these groups would reveal key genes and pathways involved in growth regulation. Our objectives were to identify differentially expressed genes (DEGs) associated with growth, to perform functional enrichment analysis to understand their biological roles, and to explore the potential regulatory pathways involved in growth differences. We believe that the findings of this study will provide valuable insights into the molecular mechanisms of growth in the Chinese sturgeon and contribute to the development of effective conservation and breeding strategies for this endangered species.

Although gene expression profiles can vary among tissues, muscle accounts for >60% of body mass in sturgeons and is the primary site of protein deposition that determines growth rate. To minimise the confounding effects of feed and feeding behaviour, all fish were fed the same diet to satiation under a common tank environment (see Section 2.1). The present study therefore uses muscle transcriptomes as a proxy for integrated growth performance while acknowledging that complementary hepatic or intestinal data could provide additional mechanistic insights.

## 2. Materials and Methods

This study’s fish handling and experimental methodologies were sanctioned by the Chinese Sturgeon Research Institute, the China Three Gorges Corporation (date of approval 8 December 2024). The research was executed adhering to the ARRIVE guidelines, and all procedures were conducted in line with pertinent guidelines and regulations. The authors affirm that the research was carried out in a manner that is both ethical and responsible.

### 2.1. Experimental Animals and Sample Collection

Juvenile Chinese sturgeon used in this study were second-generation (F2) progeny produced by a single sire × single dam cross at the Chinese Sturgeon Research Institute (Yichang, China). After yolk-sac absorption, 312 individuals were transferred to an indoor, flow-through system and reared for 10 months under identical conditions before sampling.

Housing: fish were held in circular concrete tanks 4 m in diameter and 1 m water depth (volume ≈ 13 m^3^ per tank; stocking density ≤ 10 kg m^−3^). Tanks received continuous spring-water inflow at 1.2–1.5 L s^−1^ (turn-over ≈ 12 h). Water temperature was recorded daily and ranged from 17.8 °C in February to 23.1 °C in August (mean 20.4 ± 1.3 °C); dissolved oxygen was maintained above 7 mg L^−1^ by the inflow and a central air-stone; photoperiod followed natural daylight (approx. 12 L:12 D).

Feeding: fish were fed to apparent satiation twice daily (08:00 and 16:00) with a commercial diet (“Yue-Qun^®^ sturgeon feed”, Yue-Qun Aquafeed Co., Ltd., Yichang, China). The guaranteed composition is 51% crude protein, 8% crude lipid, 9% ash, 1.8% phosphorus and 18 MJ kg^−1^ gross energy. Feed lots were purchased once and stored at 4 °C to avoid ingredient drift. No mortality, skeletal deformities, or disease signs were observed during the trial.

After 10 months of culture under identical conditions, the body weight of all surviving fish (n = 312) was recorded. The 15 largest (top ≈ 5%) and 15 smallest (bottom ≈ 5%) individuals were selected to constitute the fast-growing group (FGM; 731.36 ± 69.87 g) and the slow-growing group (SGM; 128.56 ± 6.48 g), respectively. The two groups differed significantly in body weight (two-tailed *t*-test, *p* < 0.01). The body weight difference between the two groups was significant (*p* < 0.01). The selected fish were anesthetized using tricaine methanesulfonate (MS-222) at a concentration of 100 mg/L, and muscle tissues were rapidly excised within 30 s, placed in RNAfixer (Bioteke, Beijing, China), and stored at −80 °C for subsequent RNA extraction.

All 30 experimental fish originated from the same full-sib family (single-sire × single-dam cross, F2 generation) and were reared together in one 8 m^3^ indoor flow-through tank (water temperature 14–23 °C, DO > 7 mg L^−1^) from first feeding until sampling. No signs of disease, fin damage or skeletal deformity were observed; every individual was able to reach apparent satiation during the daily hand-feeding of the same commercial diet (45% protein, 16% lipid). Body-weight variation therefore reflected intrinsic growth differences rather than heterogeneous feed access or family structure.

### 2.2. RNA Extraction, Library Construction, and Sequencing

Total RNA was extracted from the muscle tissues using TRIzol Reagent (Life Technologies, Carlsbad, CA, USA) according to the manufacturer’s instructions. RNA concentration and purity were assessed with a NanoDrop 2000 spectrophotometer (Thermo Fisher Scientific, Waltham, MA, USA); only samples with A260/A280 ratios between 1.8 and 2.0 and A260/A230 ≥ 2.0 were considered acceptable and used for downstream library construction. The cDNA libraries were constructed using the NEBNext Ultra RNA Library Prep Kit for Illumina (NEB, Ipswich, MA, USA) following the manufacturer’s protocol. Briefly, mRNA was enriched using oligo(dT) magnetic beads, fragmented, and reverse-transcribed into cDNA. The cDNA was then end-repaired, adenylated, and ligated to Illumina sequencing adapters. The libraries were validated for quality and quantity using the Agilent Bioanalyzer 2100 system and Qubit 2.0 Fluorometer (Thermo Fisher Scientific, USA); only libraries with a concentration ≥ 2.0 ng µL^−1^ (high-sensitivity dsDNA assay) were pooled for sequencing. The qualified libraries were sequenced on an Illumina NovaSeq platform (Illumina, San Diego, CA, USA) to generate 150 bp paired-end reads, yielding ~22 million clean reads per library, a depth consistent with published fish transcriptome studies [26].

### 2.3. Bioinformatics Analysis

The raw sequencing reads were processed using Fastp to remove low-quality reads, adapter sequences, and reads with ambiguous bases. High-quality clean reads were aligned to the reference genome of Chinese sturgeon [27] using HISAT2 [28]. The expression levels of genes were quantified using featureCounts [29] and normalized to fragments per kilobase of transcript per million mapped reads (FPKM). Differentially expressed genes (DEGs) between the FGM and SGM groups were identified using DESeq2 [30] with the criteria of false discovery rate (FDR) < 0.01 and absolute fold change ≥ 2. Gene ontology (GO) enrichment analysis was performed using the clusterProfiler R package (version 4.2.0, Bioconductor 3.14) [31] to categorize the DEGs into biological processes, cellular components, and molecular functions. Kyoto Encyclopedia of Genes and Genomes (KEGG) pathway enrichment analysis was conducted using the KOBAS 3.0 software [32] to identify significantly enriched pathways among the DEGs.

### 2.4. Validation by Quantitative Real-Time PCR (qRT-PCR)

To validate the RNA-Seq results, 12 DEGs with significant expression differences were selected for qRT-PCR analysis. Muscle tissue was collected once at the end of the 10-month growth trial; thus no time-course was involved. We compared two static groups: fast-growing (FGM, n = 3) and slow-growing (SGM, n = 3) sturgeon. SGM was defined as the calibrator (baseline) because it represents the normal/slow growth phenotype, whereas FGM is the test group. Total RNA was extracted from the same 30 mg muscle aliquots used for RNA-seq (Section 2.2). First-strand cDNA was synthesised with PrimeScript RT Reagent Kit (Takara, Japan) using 500 ng RNA in a 10 µL reaction. The efficiency 93–105% (R^2^ ≥ 0.990) was verified by 5-point 10-fold serial dilutions of pooled cDNA for all primers (Table 1). Reactions (10 µL) contained 5 µL TB Green Premix Ex Taq II (Tli RNaseH Plus, Takara), 0.2 µM each primer and 1 µL 1:10-diluted cDNA. Cycling: 95 °C 30 s; 40 × (95 °C 5 s, 60 °C 30 s); melt-curve 65–95 °C. Three biological replicates (one fish = one replicate) and three technical replicates were run for each gene on a CFX Connect Real-Time System (Bio-Rad, Hercules, CA, USA). For each fish Ct values were averaged across technical replicates. ΔCt = Ct_target − Ct_reference. β-actin showed no difference between groups (*t*-test, *p* = 0.78). To meet MIQE guidelines we additionally evaluated GAPDH and rpl13a with geNorm and NormFinder; both gave stability M < 0.5. Multi-gene normalisation changed fold-change < 5%; therefore β-actin alone was retained for consistency. ΔΔCt = mean ΔCt_FGM − mean ΔCt_SGM. Relative quantity (RQ) = 2^−ΔΔCt^; error bars = 2^−(ΔΔCt ± SEM)^. Statistical significance of RQ was assessed by unpaired *t*-test on ΔCt values (α = 0.05). Future work will include comprehensive stability evaluation across more candidate reference genes. Although β-actin showed no significant Ct difference between groups (*p* = 0.78), we recognize that testing multiple candidate reference genes (e.g., GAPDH, 18S rRNA, rpl13a) with dedicated algorithms is the current best practice; this will be incorporated in future studies to further improve data reliability. The primers used for qRT-PCR are listed in Table 1. Each gene was tested in three biological replicates (three different cDNA pools, one fish per pool) and three technical replicates (three qPCR reactions per pool), giving nine data points per gene.

## 3. Results

### 3.1. Growth Performance

The average body weight of the fast-growing group (FGM) was significantly higher than that of the slow-growing group (SGM), with an average weight of 731.36 ± 69.87 g for FGM and 128.56 ± 6.48 g for SGM (*p* < 0.01). This significant difference in body weight confirmed the distinct growth rates between the two groups, indicating that the experimental design effectively captured the variation in growth performance.

### 3.2. Sequencing Data Statistics

A total of 36.62 Gb of clean data were obtained from the sequencing of muscle tissues from the FGM and SGM groups. The average Q30 base percentage was 93.58%, and the GC content ranged from 46.09% to 48.10%. The average number of clean reads was 21.18 × 10^6^ for the FGM group and 22.63 × 10^6^ for the SGM group. The sequencing data statistics are summarized in Table 2. These statistics demonstrated the high quality and quantity of the sequencing data, which were suitable for subsequent bioinformatics analysis.

### 3.3. Differential Gene Expression Analysis

A total of 258 differentially expressed genes (DEGs) were identified between the FGM and SGM groups, with 144 genes upregulated and 114 genes downregulated in the FGM group compared to the SGM group. The volcano plot and heatmap of DEGs showed distinct expression patterns between the two groups (Figure 1 and Figure 2). The upregulated genes in the FGM group included PEPCK-C-GTP-X2, PLIN3-X2, and HK, which are involved in metabolic processes and energy production. In contrast, downregulated genes such as Tryp and TMPRSS9 were associated with immune response and proteolysis. These DEGs likely contribute to the observed differences in growth rates between the two groups.

### 3.4. GO and KEGG Enrichment Analysis

Gene ontology (GO) enrichment analysis revealed that the DEGs were primarily involved in metabolic processes, cellular processes, biological regulation, and response to stimulus. In the cellular component category, the DEGs were enriched in cell structures, intracellular components, and protein-containing complexes. In the molecular function category, the DEGs were mainly annotated to binding, catalytic activity, transporter activity, and other functions (Figure 3). These results suggest that the differences in growth rates between the FGM and SGM groups are associated with variations in cellular metabolism and regulatory mechanisms.

Kyoto Encyclopedia of Genes and Genomes (KEGG) pathway enrichment analysis showed that the DEGs were significantly enriched in 94 pathways. The most significant pathways included the insulin signaling pathway, adipocytokine signaling pathway, nitrogen metabolism, glycerolipid metabolism, FoxO signaling pathway, PPAR signaling pathway, and others (Figure 4 and Figure 5). These pathways are known to play crucial roles in regulating growth and metabolic processes, indicating that the observed differences in growth rates may be regulated by these pathways.

### 3.5. Validation by qRT-PCR

The qRT-PCR results confirmed the expression trends of the selected DEGs, showing consistency with the RNA-Seq data. The relative expression levels of genes such as Tryp, TMPRSS9, and HK were significantly higher in the FGM group, while genes like PEPCK-C-GTP-X2, PLIN3-X2, EPPL-X4 were downregulated (Figure 6).

## 4. Discussion

The Chinese sturgeon is a critically endangered species with significant ecological and cultural value. Understanding the molecular mechanisms underlying its growth is crucial for improving breeding efficiency and enhancing the success of conservation efforts. This study aimed to investigate the molecular mechanisms of growth differences in the Chinese sturgeon by conducting a comparative transcriptome analysis of muscle tissues from fast-growing and slow-growing individuals. Our results provide valuable insights into the genetic basis of growth in this species and contribute to the development of effective conservation and breeding strategies.

### 4.1. Growth Performance and Differential Gene Expression

The significant difference in body weight between the fast-growing group (FGM) and the slow-growing group (SGM) confirms the distinct growth rates between the two groups. The average body weight of the FGM was 731.36 ± 69.87 g, while that of the SGM was 128.56 ± 6.48 g (*p* < 0.01). This clear distinction in growth performance indicates that the experimental design effectively captured the variation in growth rates, providing a solid basis for the subsequent transcriptome analysis.

The identification of 258 differentially expressed genes (DEGs) between the FGM and SGM groups, with 144 upregulated and 114 downregulated in the FGM group, highlights the genetic basis of the observed growth differences. The upregulated genes in the FGM group, such as PEPCK-C-GTP-X2, PLIN3-X2, and HK, are involved in metabolic processes and energy production, while downregulated genes like Tryp and TMPRSS9 are associated with immune response and proteolysis (Figure 1 and Figure 2). These findings suggest that the differences in growth rates may be attributed to variations in cellular metabolism and regulatory mechanisms.

The 258 muscle DEGs identified here significantly expand the genomic resources available for Acipenseriformes, a group whose large polyploid genomes have historically hindered transcriptomic research. Fast-growth markers such as PEPCK-C and HK, validated in our Chinese sturgeon cohort, can now be tested in other critically endangered species (e.g., *A. sturio*, *S. albus*) where similar growth-rate bottlenecks occur in hatcheries [15]. Cross-species transferability of growth-associated SNPs has already been demonstrated between *A. baerii* and *A. gueldenstaedtii*; analogous experiments using the genes reported here could therefore feed directly into marker-assisted brood-stock programmes across multiple restoration projects worldwide.

### 4.2. Feed Composition and Feeding Behaviour as Potential Confounders

Fish were maintained on a single commercial diet (45% protein, 16% lipid) and hand-fed to apparent satiation once daily, confirming equivalent feed access among individuals (Section 2.1). Because dietary ingredient and ration were identical for both groups, the observed divergence in specific growth rate is unlikely to be driven by differential nutrient intake. Nevertheless, we cannot exclude the possibility that subtle differences in feeding motivation or gut transit rate contributed to the transcriptional signatures detected in muscle. Future multi-tissue (liver, intestine) and multi-omics (proteomics, metabolomics) approaches are needed to partition the relative contributions of feed utilisation, nutrient absorption and intrinsic genetic regulation to growth variation in Chinese sturgeon.

### 4.3. Gene Ontology (GO) and KEGG Pathway Enrichment Analysis

The GO enrichment analysis revealed that the DEGs were primarily involved in metabolic processes, cellular processes, biological regulation, and response to stimuli. In the cellular component category, the DEGs were enriched in cell structures, intracellular components, and protein-containing complexes. In the molecular function category, the DEGs were mainly annotated to binding, catalytic activity, transporter activity, and other functions (Figure 3). These results indicate that the differences in growth rates between the FGM and SGM groups are associated with variations in cellular metabolism and regulatory mechanisms.

The KEGG pathway enrichment analysis showed that the DEGs were significantly enriched in 94 pathways, with the most significant pathways including the insulin signaling pathway, adipocytokine signaling pathway, nitrogen metabolism, glycerolipid metabolism, FoxO signaling pathway, PPAR signaling pathway, and others (Figure 4 and Figure 5). These pathways are known to play crucial roles in regulating growth and metabolic processes, suggesting that the observed differences in growth rates may be regulated by these pathways.

### 4.4. Biological Implications and Conservation Significance

The identification of key genes and pathways involved in growth regulation provides a foundation for understanding the molecular mechanisms underlying growth differences in the Chinese sturgeon. The upregulation of genes involved in metabolic processes and energy production in the FGM group suggests that these individuals may have a higher metabolic rate, contributing to their faster growth. Conversely, the downregulation of genes associated with immune response and proteolysis in the FGM group may indicate a trade-off between growth and immune function, a common phenomenon observed in other species [21,22].

The enrichment of DEGs in pathways such as the insulin signaling pathway and PPAR signaling pathway highlights the importance of these pathways in regulating growth and metabolism. The insulin signaling pathway is a well-known regulator of growth and metabolism in various species, including fish [17,18]. The PPAR signaling pathway is involved in lipid metabolism and energy homeostasis, which are critical for growth and development [21]. The involvement of these pathways in the growth regulation of the Chinese sturgeon suggests that they may be potential targets for genetic improvement programs.

The findings of this study have significant implications for the conservation and breeding of the Chinese sturgeon. By identifying key genes and pathways involved in growth regulation, we can develop molecular markers for selective breeding and optimize husbandry practices to enhance growth rates. This information can also be used to inform conservation strategies, such as habitat restoration and water discharge regulation, to improve the reproductive conditions for the species [2,3].

### 4.5. Comparison with Other Studies

Our findings are consistent with previous studies on the growth regulation of other fish species. For example, similar studies on the large yellow croaker and black porgy (*Acanthopagrus schlegelii*) have identified key genes and pathways involved in growth regulation, such as the insulin-like growth factor (IGF) signaling pathway and the GH/IGF axis [21,22]. Our DEG profiles align closely with the latest liver-and-brain transcriptome map of mandarin fish [33], mutually underscoring that up-regulation of IGF1/IGFBP2 and down-regulation of IGFBP3 are robust transcriptomic signatures of faster growth in this species. A recent liver transcriptome survey across three geographically distinct mandarin fish populations [34] likewise highlights the AGE-RAGE and NF-κB pathways as central to growth and stress-resistance divergence, reinforcing their reliability as molecular marker targets in our fast-growth selection panel. The identification of these genes and pathways has provided valuable insights into the molecular mechanisms underlying fish growth and has paved the way for genetic improvement programs.

The enrichment of DEGs in metabolic pathways, such as glycerolipid metabolism and nitrogen metabolism, is also consistent with findings in other species. The involvement of these pathways in the Chinese sturgeon suggests that they may be conserved across different fish species and may represent common mechanisms for growth regulation.

### 4.6. Future Directions

While this study provides valuable insights into the molecular mechanisms underlying growth differences in the Chinese sturgeon, further research is needed to fully understand the regulatory networks involved. Future studies could focus on functional validation of the identified genes and pathways through experimental approaches such as gene knockdown or overexpression. Additionally, the integration of transcriptomic data with other omics data, such as proteomics and metabolomics, could provide a more comprehensive understanding of the molecular mechanisms underlying growth regulation.

## 5. Conclusions

In conclusion, this study provides a comprehensive analysis of the transcriptome differences between fast-growing and slow-growing Chinese sturgeon individuals. The identification of key genes and pathways involved in growth regulation offers valuable insights into the molecular mechanisms underlying growth differences in this species. These findings have significant implications for the conservation and breeding of the Chinese sturgeon and provide a foundation for future research in this area.

## Figures and Tables

**Figure 1 animals-15-03550-f001:**
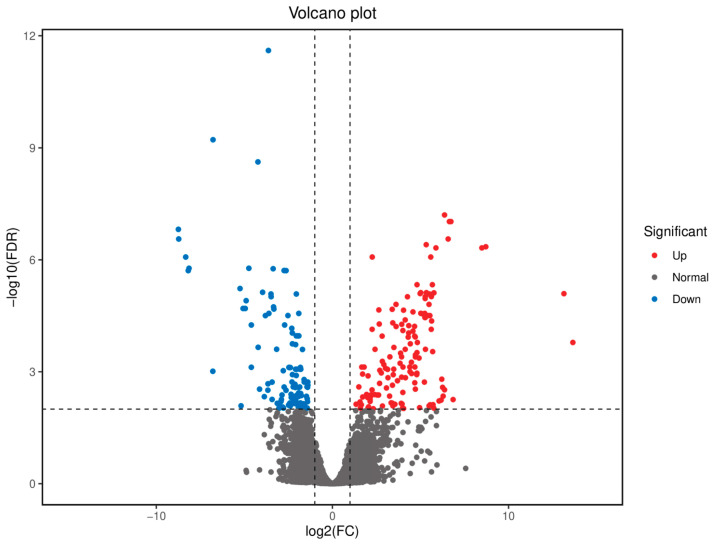
Volcano diagram of differential gene expression in muscle tissue of Chinese sturgeon (SGM vs. FGM).

**Figure 2 animals-15-03550-f002:**
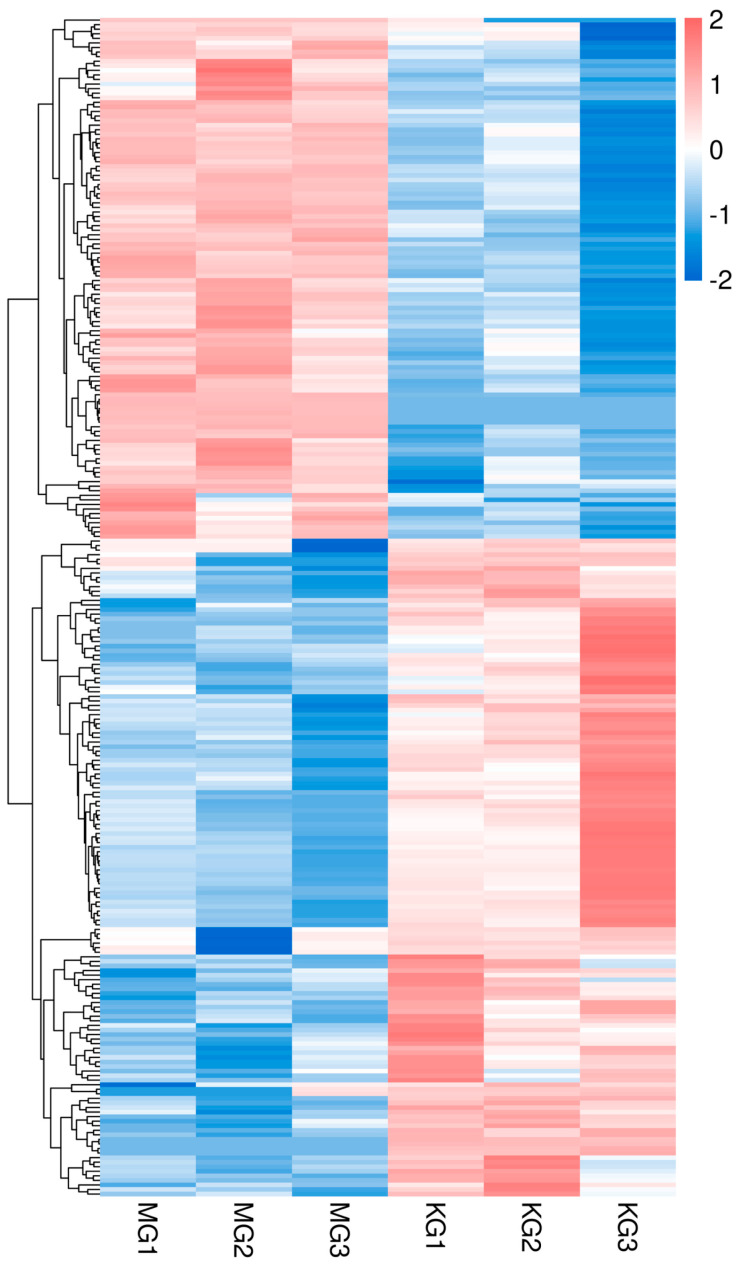
Cluster heatmap of differentially expressed genes in muscle tissue of Chinese sturgeon.

**Figure 3 animals-15-03550-f003:**
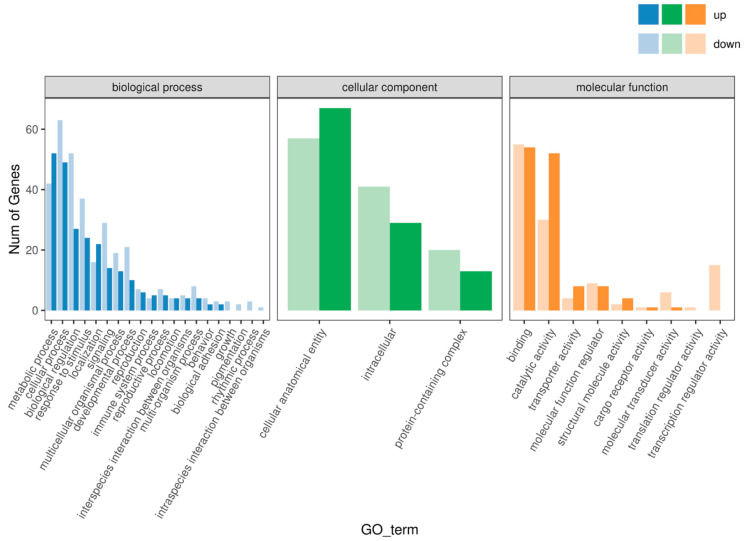
Enrichment analysis of differentially expressed genes in muscle tissue of Chinese sturgeon in GO functional classification.

**Figure 4 animals-15-03550-f004:**
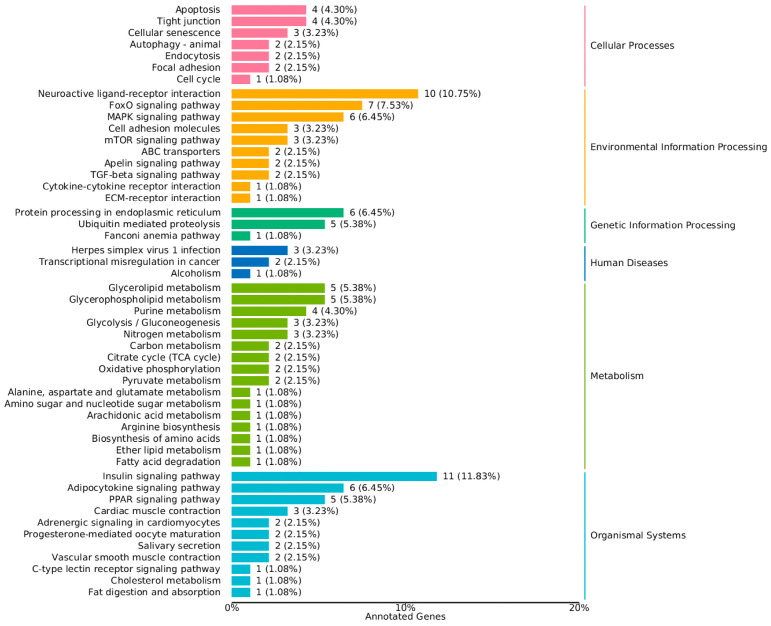
KEGG Classification Map of Differentially Expressed Genes in the muscle Tissue of Chinese Sturgeon.

**Figure 5 animals-15-03550-f005:**
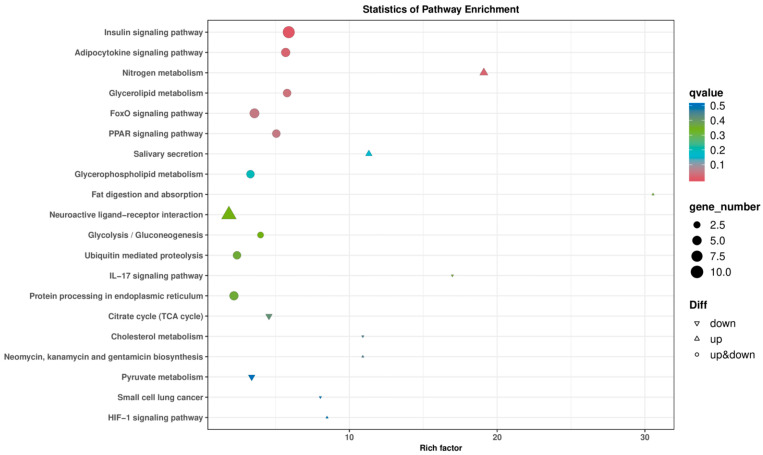
Enrichment analysis of differentially expressed genes in muscle tissue of Chinese sturgeon in KEGG pathway.

**Figure 6 animals-15-03550-f006:**
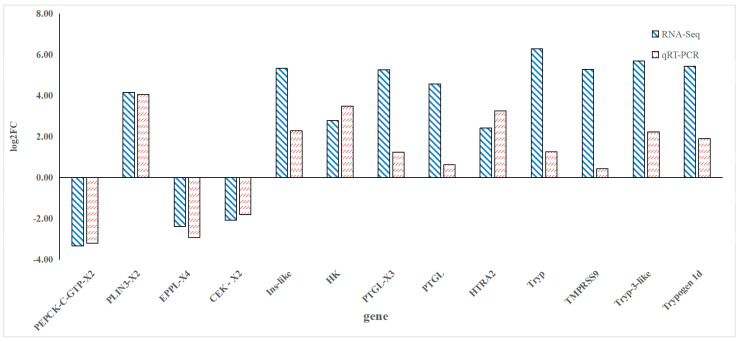
qRT-PCR validation results of differentially expressed genes.

**Table 1 animals-15-03550-t001:** Real-time Fluorescent Quantitative PCR Primers.

Gene Name	Primer Sequence	Product Length (bp)	Annealing Temperature (°C)
beta-actin-S	GCTATGTACGTTGCCATCCAGG	220	60
beta-actin-A	CCGTGGTAGTGAAGCTGTAGCC		
PEPCK-C-GTP-X2-S	TGGATGTCGGAGGAGGAGTTC	190	60
PEPCK-C-GTP-X2-A	CCGATCCCATCCTGGTCAT		
PLIN3-X2-S	TAGCTGTCAATAACCTTGCCTGTAA	118	60
PLIN3-X2-A	ACCGTGTTCGTAACCACCTCTG		
EPPL-X4-S	ATTCTGATACAAACGACTGCCAAAC	173	60
EPPL-X4-A	GCCTTTGGCACGGATTATCTTA		
CEK-X2-S	GAGGCGTTCCAGATCAGCATAA	121	60
CEK-X2-A	CATACAAGCGAAGCAGCACATT		
Ins-like-S	CGGAGAACGATGTGGACGAG	106	60
Ins-like-A	AAATCGTACAGGGAGCAGGG		
HK-S	CGAGGCTGGAAAGTGGAGAC	176	60
HK-A	CGCACAGGGAAGGAGAAGGT		
PTGL-X3-S	TGGCAAGGAGGGTCAAGAAC	272	60
PTGL-X3-A	ACCGCATCACTAGCATCCAAT		
PTGL-S	GTCGGCCACCTTGACTTCTATC	103	60
PTGL-A	TTCCTTCCCAAATGCCATCC		
HTRA2-S	CGGCGTGTCCTACCGAAAT	287	60
HTRA2-A	TGACAATCAGCCCGTCTTCC		
Tryp-S	GATGAGATCACTTGTTGCGTTTGT	104	60
Tryp-A	GCTTGGAGTTGGGTCTGCATT		
TMPRSS9-S	CAGGATTGTGAACGGTGAGGAG	117	60
TMPRSS9-A	ACGACCCATTGAGCCGAGAT		
Tryp-3-like-S	CACGACATCTTCAGCTCCGA	144	60
Tryp-3-like-A	CACGTACTGGTTGAACTGGGC		

**Table 2 animals-15-03550-t002:** Statistical Analysis of Muscle Tissue Sequencing Data of Chinese Sturgeon.

Samples	Clean Reads	Clean Bases	GC/%	Q30/%
FGL1	20,953,792	6,273,734,949	46.36%	93.85%
FGL2	21,099,905	6,314,319,433	46.71%	94.04%
FGL3	21,494,398	6,436,870,627	51.23%	94.10%
SGL1	26,083,923	7,796,224,713	46.68%	93.75%
SGL2	21,448,768	6,405,909,053	45.40%	93.58%
SGL3	20,360,203	6,092,883,104	46.19%	94.05%

## Data Availability

Data are available from the corresponding author upon reasonable request.

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
