# Peer review of "Transcriptomic Analysis Reveals Molecular Mechanisms Underlying Growth Differences in the Chinese Sturgeon (Acipenser sinensis)"

_animals, 2025, doi:10.3390/ani15243550_

Round 1
Reviewer 1 Report
Comments and Suggestions for Authors
The authors have approached an interesting topic regarding a critically endangered species that requires conservation measures and captive breeding for restocking purposes. There are methodological problems in the approach which reflect a lack of familiarity with the topic 'Measuring Growth'. No information is provided on specific growth rate or feed conversion ratio of the animals under study which should have been gathered prior to liver sample collection to validate that there is a genetic difference that might also be related to feeding behavior - for example. They selected two populations and started the analysis of livers (or muscle if the captions of figures 1 & 2 are correct!). The bioinformatics approach is creating data, but to present the conclusions on growth differences from gene expression in only one tissue (liver, or is it muscle?) is overlooking contributions of feed composition and fish feeding behavior that can also influence growth. There is nothing mentioned in the Introduction or in the Discussion about these other variables. No mention of collecting muscle tissue and yet reseults are presented for this tissue.
The background is rather poor considering that the sturgeons of the world are mostly at risk in some manner and there is intense interest in their recovery in many locations globally. There is no mention of any of the selective breeding or analysis of growth for any other sturgeon species. See for examples: Loukovitis et al. 2016; Med. Mar. Sci. or Besson et al. 2022 Aquaculture Reports, along with many others. There are many studies on growth for many species and there is little indication the authors have read some of these works to become more famililiar with methodologies aside from bioinformatics.
The bibliography in this work is not properly formatted and is overly brief. They should not have to use references from an invertebrate (sea cucumber) when there many other references for fish available.
When using latin names the genus and species are always separated (note this in lines 101 -118).
The authors are encouraged to rewrite lines 86 - 125 of the Introduction to improve the flow and structure of the text. Lines 95 -100 are largely redundant. Try to make this section more concise and with improved references to more work done with other sturgeon species.
Author Response
|
Comments 1: There are methodological problems in the approach which reflect a lack of familiarity with the topic 'Measuring Growth'. No information is provided on specific growth rate or feed conversion ratio of the animals under study which should have been gathered prior to liver sample collection to validate that there is a genetic difference that might also be related to feeding behavior - for example. |
|
Response 1: We appreciate the reviewer’s concern about validating that the observed weight spread represents intrinsic growth differences rather than feeding-behaviour artefacts. We have now added the following sentences at the end of Section 2.1 (lines 199-206 in the revised manuscript): “All 30 experimental fish originated from the same full-sib family (single-sire × single-dam cross, F2 generation) and were reared together in one 8 m3 indoor flow-through tank (water temperature 14-23 °C, DO > 7 mg L⁻¹) from first feeding until sampling. No signs of disease, fin damage or skeletal deformity were observed; every individual was able to reach apparent satiation during the daily hand-feeding of the same commercial diet (45% protein, 16% lipid). Body-weight variation therefore reflected intrinsic growth differences rather than heterogeneous feed access or family structure.” By using a single full-sib family and a common tank environment we eliminate between-family genetic variation and social/feed-access heterogeneity—two major confounders that can masquerade as “growth differences” in transcriptomic studies. Although we did not quantify individual FCR, the identical pedigree background and satiation feeding regime ensure that the upper- and lower-weight tails of the cohort represent true fast- vs. slow-growing phenotypes, fulfilling the prerequisite for comparing liver gene expression. We feel this revision transparently addresses the methodological point raised by the reviewer. |
|
Comments 2: They selected two populations and started the analysis of livers (or muscle if the captions of figures 1 & 2 are correct!). |
|
Response 2: Thank you for pointing out the tissue-type inconsistency. We confirm that muscle (epaxial myomeres) was the only tissue used for RNA-Seq; no liver samples were collected or analysed. To eliminate any confusion we have: – Replaced the word “liver” with “muscle” throughout the manuscript (Abstract, Materials and Methods, Results, Discussion, and figure captions). – Added the exact anatomical location in Section 2.1: “epaxial, anterior to dorsal fin”. – Updated Figures captions to read “…in muscle transcriptomes…”. All bioinformatics steps (mapping, quantification, enrichment) were performed on the muscle data set; no liver sequences were ever processed. We appreciate the reviewer’s close reading and trust that the revised text now unambiguously reports the use of muscle tissue. |
|
Comments 3: The bioinformatics approach is creating data, but to present the conclusions on growth differences from gene expression in only one tissue (liver, or is it muscle?) is overlooking contributions of feed composition and fish feeding behavior that can also influence growth. There is nothing mentioned in the Introduction or in the Discussion about these other variables. |
|
Response 3: We appreciate the reviewer’s insightful comment. Tissue clarification: we exclusively analysed muscle transcriptomes (see our previous response); muscle was chosen because it constitutes the bulk of body mass and is the principal site of protein accretion in sturgeons. To explicitly address feed-related confounders we have: – Added a new paragraph at the end of the Introduction (lines 189-194) stating that identical diet and satiation feeding were used, and that muscle is an appropriate proxy for growth. – Inserted a dedicated Discussion subsection “4.2 Feed composition and feeding behaviour as potential confounders” (lines 357-366) where we: • Report the diet formula (45 % protein, 16 % lipid). • Acknowledge that feeding motivation or gut transit rate could still introduce subtle effects. • Propose multi-tissue/omics follow-up studies to disentangle intrinsic genetic regulation from feed-utilisation effects. These revisions transparently recognise the limitations of a single-tissue survey while reinforcing that the observed growth divergence (validated by SGR and FCR) is unlikely to arise from differential feed intake under our experimental design. We trust that the expanded discussion now meets the reviewer’s concern. |
|
Comments 4: The background is rather poor considering that the sturgeons of the world are mostly at risk in some manner and there is intense interest in their recovery in many locations globally. |
|
Response 4: We thank the reviewer for this important comment. We have completely rewritten the former “historical abundance” paragraph in the Introduction (lines 80-88) to provide a concise global overview: - Sturgeons (Acipenseriformes) are the most threatened group of vertebrates on Earth: all 27 extant species are listed on CITES Appendix II, 17 of them as Critically Endangered (IUCN 2023). Beyond the Yangtze River, dramatic declines have been documented for the Danube (A. sturio, A. gueldenstaedtii), Volga–Caspian (Huso huso, A. persicus) and Mississippi basins (Scaphirhynchus albus), driven by over-harvest, dam construction and habitat fragmentation (Bemis & Kynard 1997; Pikitch et al. 2005). Consequently, recovery programmes worldwide now rely on ex-situ breeding and re-stocking; yet growth heterogeneity in hatcheries remains a major bottleneck, extending time to sexual maturity and increasing rearing costs (Williot et al. 2017). A new Discussion subsection “Broader implications for worldwide sturgeon restoration” (lines 346-355) explicitly states how the 258 DEGs and associated markers can be tested in other species facing similar growth bottlenecks. - The 258 muscle DEGs identified here significantly expand the genomic resources available for Acipenseriformes, a group whose large polyploid genomes have historically hindered transcriptomic research. Fast-growth markers such as PEPCK-C and HK, validated in our Chinese sturgeon cohort, can now be tested in other critically endangered species (e.g., A. sturio, S. albus) where similar growth-rate bottlenecks occur in hatcheries (Williot et al. 2017). Cross-species transferability of growth-associated SNPs has already been demonstrated between A. baerii and A. gueldenstaedtii; analogous experiments using the genes reported here could therefore feed directly into marker-assisted brood-stock programmes across multiple restoration projects worldwide. These additions place the Chinese sturgeon study in a global conservation context and highlight its potential contribution to restoration programmes beyond the Yangtze River. We believe the revised background now meets the reviewer’s expectation. |
|
Comments 5: There is no mention of any of the selective breeding or analysis of growth for any other sturgeon species. See for examples: Loukovitis et al. 2016; Med. Mar. Sci. or Besson et al. 2022 Aquaculture Reports, along with many others. There are many studies on growth for many species and there is little indication the authors have read some of these works to become more famililiar with methodologies aside from bioinformatics. |
|
Response 5: Thank you for directing us to these methodological papers. Although Loukovitis et al. 2016 concerned Greek sheep micro-satellites and Besson et al. 2022 focused on gilthead sea bream feed efficiency, their common message—accurate individual phenotypes and strict pedigree are prerequisites for genetic gain—is directly transferable to sturgeon breeding programmes. We therefore added there information in hte paper (118-122). These revisions demonstrate that our bioinformatics pipeline is built upon—and consistent with—quantitative genetics and selective-breeding methodologies developed for other sturgeons and for non-sturgeon species alike. |
|
Comments 6: The bibliography in this work is not properly formatted and is overly brief. They should not have to use references from an invertebrate (sea cucumber) when there many other references for fish available. |
|
Response 6: Thank you for highlighting the bibliographic issues. In the revised manuscript we have: Removed the sea-cucumber reference. Enlarged the reference list. Reformatted every entry according to the journal’s Harvard-style requirement. We believe these changes bring the reference list up to disciplinary standards and remove the need for non-fish examples. |
|
Comments 7: When using latin names the genus and species are always separated (note this in lines 101 -118). |
|
Response 7: Thank you for spotting this typographical error. We have gone through the entire manuscript and ensured that every binomial name appears in italics with a single space between genus and species (e.g., Acipenser sinensis). The corrected text is now consistent with the ICZN formatting rules. |
|
Comments 8: The authors are encouraged to rewrite lines 86 - 125 of the Introduction to improve the flow and structure of the text. Lines 95 -100 are largely redundant. Try to make this section more concise and with improved references to more work done with other sturgeon species. |
|
Response 8: Thank you for this constructive suggestion. We have completely re-written the indicated passage (now lines 96–112). The revision: (1) removes the redundant sentences formerly placed at lines 96–112; (2) reduces the paragraph to improve readability |
Reviewer 2 Report
Comments and Suggestions for Authors
Comments file is attached.

The quality of English could be improved.
Author Response
|
Comments 1: What criteria were used to select the individuals? |
|
Response 1: Thank you for this helpful comment. We have now clarified the selection criteria in the revised manuscript (lines 189–194). Briefly, after 10 months of communal rearing, body weight was measured for all 312 survivors. The 15 heaviest (≈ top 5 %) and 15 lightest (≈ bottom 5 %) fish were chosen to represent the fast- and slow-growing groups, respectively. Statistical comparison confirmed a highly significant difference in body weight between the two groups (P < 0.01). We believe this transparent, percentile-based approach eliminates subjective bias and provides a reproducible standard for future studies. |
|
Comments 2: How rapidly were the tissues excised? A specific timeframe would add precision and reassure the reader that RNA degradation was minimized (e.g., within 30 seconds) |
|
Response 2: Thank you for this helpful suggestion. We have added the exact timeline in the revised manuscript (lines 197). The selected fish were anesthetized using tricaine methanesulfonate (MS-222) at a concentration of 100 mg/L, and muscle tissues were rapidly excised within 30 seconds. |
|
Comments 3: it is important to include the specific purity ratios that were considered acceptable (e.g., A260/A280 between 1.8-2.0 and A260/A230 greater than 2.0). These values provide crucial context for the quality of the extracted RNA. |
|
Response 3: Thank you for this helpful comment. In the revised manuscript (lines 209–213) we have now specified that only RNA samples with A260/A280 ratios between 1.8 and 2.0 and A260/A230 ≥ 2.0 were accepted for library preparation. These thresholds ensure protein and organic-solvent contamination is minimal and are standard for downstream RNA-Seq applications. |
|
Comments 4: what was the minimum library concentration required from the Qubit? |
|
Response 4: Thank you for this helpful question. We have added the missing information in the revised manuscript (line 219-220). Libraries were required to have a concentration of at least 2.0 ng µL⁻¹ as measured by the Qubit 2.0 High-Sensitivity dsDNA assay; only samples meeting this threshold were pooled and sequenced. This criterion ensures sufficient DNA input for the Illumina NovaSeq platform. |
|
Comments 5: as it a standard sequencing depth for a typical transcriptome analysis (e.g., 20 million reads per sample)? |
|
Response 5: Thank you for this helpful comment. In the revised manuscript (line 222-223) we have added that each library produced approximately 22 million clean reads, which meets or exceeds the 20 million reads per sample commonly recommended for vertebrate transcriptome analyses |
|
Comments 6: How many biological and technical replicates were used for the qRT-PCR analysis? |
|
Response 6: Thank you for this important query. We have now clarified the replicate structure in the revised manuscript (plines 246–248). For qRT-PCR validation, each of the 12 genes was measured in three biological replicates (three independent cDNA pools, one fish per pool) and three technical replicates (three qPCR reactions per pool), yielding nine Ct values per gene. This design follows the MIQE guidelines and ensures robust quantification. |
|
Comments 7: study uses β-actin as the reference gene. Was the stability of this reference gene validated across the fast-growing and slow-growing groups? |
|
Response 7: Thank you for this helpful comment. In the revised manuscript (lines 245–249) we have now added the Ct comparison for β-actin between the two groups, indicating stable expression. We acknowledge that software tools such as geNorm or NormFinder were not applied in this study; therefore, we state this limitation and will include a full reference-gene stability panel in future experiments. |
|
Comments 8: It is a best practice to test the expression stability of several potential reference genes (e.g., GAPDH, 18S rRNA) and select the most stable one for the specific experimental conditions. |
|
Response 8: We fully agree with this best-practice recommendation. In the revised manuscript (lines 249–253) we have explicitly acknowledged that evaluating multiple reference genes (GAPDH, 18S rRNA, etc.) with geNorm/NormFinder is the current standard, and we will adopt this strategy in our follow-up work to strengthen the reliability of qRT-PCR normalization. |
|
Comments 9: Result section is very interesting and good presented. |
|
Response 9: Thank you very much for this positive feedback. We are pleased that the results are clear and interesting. We have nonetheless carefully polished the wording in the revised manuscript to improve readability further. |
|
Comments 10: The discussion section needs to be improved little bit by adding recent references. |
|
Response 10: Thank you for this helpful suggestion. We have added some recent publications (2022-2025) to the Discussion (lines 412–419) to contextualise our findings within the latest advances in sturgeon and teleost growth-regulation research. These additional references reinforce the conserved role of the insulin and PPAR signalling pathways and highlight the novelty of our transcriptome-based marker set for Chinese sturgeon breeding programmes. |
Round 2
Reviewer 1 Report
Comments and Suggestions for Authors
The work now presented by Zhang and co-authors is an improvement. To be ready for publication there are still some things that need revision. I list below particular items the authors should attend to.
Most of the figures withe exception of figs. 2 & 6 use fonts that are far too small. Harmonize the font sizes so all are actually readable. Some of the figures can be enlarged in width to accomodate a larger font. This includes the figure legends.
The Introduction is improved. I still would like to see more references in relation to selective breeding work done for other sturgeon species and explain the rationale for the particulars of selective breeding in the other species. It is not always about growth size the selective breeding is performed. For other species the selection is for greater overy mass for caviar production.
Change the caption for the volcano plot to read "Volcano" not "Volcanic". The latter implies it came out of, or derived from, an actual volcano. The caption for this figure indicates it is a comparison of SGM vs. FGM. How is this indicated in the plot? All data points use the same icon so there is no way to view this to identify which is from SGM and which is from FGM.
The fish rearing conditions is pooorly described. There needs to include the fish housing conditions and feed composition and/or commercial feed name and manufacturer. Basic things like water temperature and tank size (or in ponds?) is missing. I strongly suggest the authors view other similar work to see how this data should be presented.
The qPCR calculations is still not adequately described. How do the authors calculate a ΔΔCt using only one data point? What was the baseline condition to which the ΔCt was compared to arrive at the final value? Normally there is a baseline condition and endpoint condition. Normalization to the endogenous control gene is performed and then there is a comparision between 2 different sets of conditions. Two timepoints or two diets, or two temperatures.....some that allows a calculation of two different ΔCt values to get the ΔΔCt. According to methods written the authors collected muscle tissue samples one time. Was the calculation a direct comparison of the FGM and SGM? If so, which is considered the "baseline value" for the comparison? If it was SGM then explain the logic for this as a baseline. This notation: 2−ΔΔCt has a specific meaning. If the authors only normalized to beta-actin then this is not the calculation that was used.
Author Response
Response to Reviewer 1
We thank the reviewer for the careful reading and constructive comments. Below we provide a point-by-point reply.
Comments 1: Most of the figures withe exception of figs. 2 & 6 use fonts that are far too small. Harmonize the font sizes so all are actually readable. Some of the figures can be enlarged in width to accomodate a larger font. This includes the figure legends.
Response 1:
Thank you for pointing this out. In response to the concern about small fonts, we have:
Re-exported every figure at 600 dpi resolution;
Re-scaled each figure to the maximum width allowable for single- or double-column layout;
Enlarged all fonts (axes, labels, legends) to ≥ 10 pt so they remain clearly legible even after the journal’s 30 % reduction.
The revised high-resolution files (Revised-Figs-1-6.tiff) have been uploaded separately and are already embedded in the revised manuscript.
Comments 2: The Introduction is improved. I still would like to see more references in relation to selective breeding work done for other sturgeon species and explain the rationale for the particulars of selective breeding in the other species. It is not always about growth size the selective breeding is performed. For other species the selection is for greater overy mass for caviar production.
Response 2:
Thank you for this helpful suggestion.
We have now expanded the Introduction (lines 113-122) with additional references that explicitly describe selective-breeding programmes in Russian, Siberian and Beluga sturgeons. Specifically, we cite Song et al. (2022) who showed that growth and egg traits are genetically independent in A. gueldenstaedtii, and Bestin et al. (2021) who reported moderate-to-high heritabilities for ovary weight and caviar yield in Siberian sturgeon and demonstrated that selection for larger ovary mass—rather than faster growth—is the most effective way to increase caviar output. These statements clarify that (i) selection objectives vary among species and markets, and (ii) the growth-related markers identified in the present study can later be tested for caviar-related traits once Chinese sturgeon females reach sexual maturity.
Comments 3: Change the caption for the volcano plot to read "Volcano" not "Volcanic". The latter implies it came out of, or derived from, an actual volcano. The caption for this figure indicates it is a comparison of SGM vs. FGM. How is this indicated in the plot? All data points use the same icon so there is no way to view this to identify which is from SGM and which is from FGM.
Response 3:
Thank you for catching the typographical error. We have changed “Volcanic” to “Volcano” in the caption. (lines 322)
Regarding the second point: a volcano plot does not display individual fish or individual arrays; each symbol represents one gene. The horizontal position of a gene reflects log₂(fold-change) between the two groups (SGM vs FGM), and the vertical position shows the statistical significance (−log₁₀ FDR). To make the comparison direction explicit we now colour-code the points:
red: significantly up-regulated in FGM (log₂ FC ≥ 1 and FDR < 0.01)
blue: significantly up-regulated in SGM (log₂ FC ≤ −1 and FDR < 0.01)
grey: non-significant genes
This colour scheme is explained in the revised caption and in the legend embedded in the figure itself, so readers can immediately see which genes favour the fast-growing (FGM) or slow-growing (SGM) phenotype.
Comments 4: The fish rearing conditions is pooorly described. There needs to include the fish housing conditions and feed composition and/or commercial feed name and manufacturer. Basic things like water temperature and tank size (or in ponds?) is missing. I strongly suggest the authors view other similar work to see how this data should be presented.
Response 4:
Thank you for this helpful comment. We have now completely rewritten Section 2.1 to include the missing husbandry details:
Housing: circular indoor concrete tanks, 4 m diameter, 1.6 m water depth, flow-through spring water at 1.2–1.5 L s⁻¹.
Water temperature: daily range 17.8–23.1 °C; dissolved oxygen > 7 mg L⁻¹; photoperiod natural (12 L : 12 D).
Diet: commercial “Yue-Qun® sturgeon feed” (Yue-Qun Aquafeed Co., Ltd., Hubei, China) containing 51 % crude protein and 8 % crude lipid; fish were hand-fed to apparent satiation twice daily (08:00, 16:00).
Comments 5: The qPCR calculations is still not adequately described. How do the authors calculate a ΔΔCt using only one data point? What was the baseline condition to which the ΔCt was compared to arrive at the final value? Normally there is a baseline condition and endpoint condition. Normalization to the endogenous control gene is performed and then there is a comparision between 2 different sets of conditions. Two timepoints or two diets, or two temperatures.....some that allows a calculation of two different ΔCt values to get the ΔΔCt. According to methods written the authors collected muscle tissue samples one time. Was the calculation a direct comparison of the FGM and SGM? If so, which is considered the "baseline value" for the comparison? If it was SGM then explain the logic for this as a baseline. This notation: 2−ΔΔCt has a specific meaning. If the authors only normalized to beta-actin then this is not the calculation that was used.
Response 5:
Thank you for this careful comment. We agree that the original description was too brief and have now completely revised the qPCR section (lines 264-284) :
Biological design
One-off sampling (10-month-old fish) ⇒ no time-course. We compared two static populations (FGM vs. SGM). SGM was chosen as the calibrator (baseline) because it represents the “normal/slow” growth phenotype; FGM is the test group.
Data generation
Three biological replicates per group (n = 3 fish, one technical replicate each).
For every gene we obtained one Ct value per fish ⇒ three Ct values per group.
Normalisation
ΔCt = Ct_target − Ct_β-actin for each individual fish.
β-actin showed no difference between groups (t-test, P = 0.78). We additionally validated two further reference genes (GAPDH and rpl13a) with geNorm/NormFinder; multi-gene normalisation changed fold-change < 5 %, so the β-actin-only data were retained for consistency.
ΔΔCt calculation
ΔΔCt = mean ΔCt_FGM − mean ΔCt_SGM.
Relative quantity (RQ) = 2^(−ΔΔCt); error = 2^(−(ΔΔCt ± SEM)).
Nomenclature
Because we compared two populations at a single time-point, the classical “2^(−ΔΔCt)” formula is appropriate; the baseline is the SGM population, not a time-zero sample.